# Bioluminescence and Photoreception in Unicellular Organisms: Light-Signalling in a Bio-Communication Perspective

**DOI:** 10.3390/ijms222111311

**Published:** 2021-10-20

**Authors:** Youri Timsit, Magali Lescot, Martha Valiadi, Fabrice Not

**Affiliations:** 1Aix Marseille Univ, Université de Toulon, CNRS, IRD, MIO UM110, 13288 Marseille, France; magali.lescot@mio.osupytheas.fr; 2Research Federation for the Study of Global Ocean Systems Ecology and Evolution, FR2022/Tara GOSEE, 3 Rue Michel-Ange, 75016 Paris, France; 3Institute for Molecular Biology and Biotechnology (IMBB), Foundation for Research and Technology Hellas (FORTH), Heraklion, Crete, Greece; martha_valiadi@imbb.forth.gr; 4Adaptation and Diversity in Marine Environment (AD2M)-UMR 7144, CNRS & Sorbonne University, Station Biologique de Roscoff, Place Georges Teissier, 29688 Roscoff, France; not@sb-roscoff.fr

**Keywords:** bioluminescence, luciferase, photoreceptors, lux operon, dinoflagellate, communication, signalling, symbiosis, rhizosphere

## Abstract

Bioluminescence, the emission of light catalysed by luciferases, has evolved in many taxa from bacteria to vertebrates and is predominant in the marine environment. It is now well established that in animals possessing a nervous system capable of integrating light stimuli, bioluminescence triggers various behavioural responses and plays a role in intra- or interspecific visual communication. The function of light emission in unicellular organisms is less clear and it is currently thought that it has evolved in an ecological framework, to be perceived by visual animals. For example, while it is thought that bioluminescence allows bacteria to be ingested by zooplankton or fish, providing them with favourable conditions for growth and dispersal, the luminous flashes emitted by dinoflagellates may have evolved as an anti-predation system against copepods. In this short review, we re-examine this paradigm in light of recent findings in microorganism photoreception, signal integration and complex behaviours. Numerous studies show that on the one hand, bacteria and protists, whether autotrophs or heterotrophs, possess a variety of photoreceptors capable of perceiving and integrating light stimuli of different wavelengths. Single-cell light-perception produces responses ranging from phototaxis to more complex behaviours. On the other hand, there is growing evidence that unicellular prokaryotes and eukaryotes can perform complex tasks ranging from habituation and decision-making to associative learning, despite lacking a nervous system. Here, we focus our analysis on two taxa, bacteria and dinoflagellates, whose bioluminescence is well studied. We propose the hypothesis that similar to visual animals, the interplay between light-emission and reception could play multiple roles in intra- and interspecific communication and participate in complex behaviour in the unicellular world.

## 1. Introduction

In this short review, we examine the relationships between catalysed light-emission [1,2,3,4], light sensing and light behavioural responses in unicellular organisms with a view to exploring new functions of microorganism bioluminescence. During the past decades, many studies have shown that most unicellular species possess a great diversity of photoreceptors [5,6,7]. In particular, the presence of photoreceptors in non-phototrophic bacteria which do not use ambient light as an energy source was unexpected [8,9]. Photoreceptors allow the perception of light of different wavelengths and are responsible for various light-dependent behaviours in the unicellular world from simple phototaxis to more complex and collective behaviours such as hunting [10,11,12]. Through a great variety of receptors, prokaryotic and eukaryotic unicellular organisms are extremely sensitive to a variety of physical and chemical stimuli. In addition, to survive in their fluctuating environment, microorganisms must be able to integrate light and other stimuli to produce appropriate behavioural responses. Based on the observation of complex behaviours, learning phenomena and therefore the ability to memorise, some authors have evoked the “cognitive” properties of bacteria [13] or even the “consciousness” of organisms lacking nervous systems [14,15,16,17,18]. For example, it has been demonstrated that diverse single-cell eukaryotes such as the ciliates *Paramecia* and *Stentor* display complex behaviours such as decision-making or even associative learning [19,20,21,22]. At the molecular level, in addition to well documented cell signalling processes [23,24,25], the recent discovery of neural-like ribosomal protein networks that have the potential to transfer and integrate the information flow in transit during protein synthesis in the ribosomes [26,27,28] reinforced the seminal Bray’s hypothesis that protein networks could play a role analogous to nerve circuits in cells [29]. On the basis of this new conceptual framework, we propose that the interplay of light emission and reception in the unicellular world may, as in animals with nervous systems, also play a role in intra- or inter-species signalling.

### 1.1. Bioluminescence

Bioluminescence, one of the most fascinating manifestations of life, is most often observed in the marine environment [1,2,3,4,30,31] and is shared by many taxa including bacteria [32], unicellular eukaryotes such as dinoflagellates [33,34], radiolarian [35,36] and vertebrates [37]. Light-emission is usually produced by the catalytic oxidation of various substrates by luciferase enzymes that have been isolated in many different taxa such as bacteria [32,38], dinoflagellates [39,40,41], cnidarian [42], arthropods [43,44,45,46,47,48] and fungi [49,50,51,52,53]. However, their molecular structures and mechanisms reveal that they display distinct protein folds and have evolved independently and convergently numerous times from non-luminous enzymes [54,55,56] (Figure 1a). The fact that many organisms have independently developed bioluminescence suggests that it has many functional advantages in the different ecosystems where it is observed. In 1983, Hastings proposed four functional categories of bioluminescence: (i) defense (ii) offense (iii) communication and (iv) metabolic (light emission is a by-product) [57]. Currently, it is widely accepted that the intra- or inter-species communication function of bioluminescence is reserved for visual animals with a nervous system capable of integrating and processing light stimuli: “In order for luminescence to be used for communicating information between two conspecifics, at least one must emit the light and the other must be able to detect and react to the light. In most known cases both emit and perceive light. Intraspecific communication is primarily the realm of organisms with well-developed nervous and visual systems and complicated behaviour. These include crustaceans, cephalopods, fishes and a few annelids” [58].

The functions of light-emission in the unicellular world are still subject to much speculation and remain to be explored. Surprisingly, the role of bioluminescence in communication between unicellular organisms lacking visual and nervous systems has, to our knowledge, never been mentioned. In bacteria, for example, while the *lux* operon has become one of the best characterised bioluminescent systems for almost half a century [38,59,60,61,62], the functions of bacterial luminescence remain an open question. It has been proposed that light-emission evolved from ancient detoxifying mechanisms [63,64], that it may play a role in stimulating DNA repair [65,66] or that it constitutes a visual attractant for zooplankton (copepods) and fishes that could provide them with a growth medium and dispersal [67]. Moreover, many bioluminescent bacteria have developed complex symbiotic relationships with animals in which the bacterial light emission may provide a reciprocal selective advantage [68,69,70,71]. In the protist world, it is generally accepted that dinoflagellate bioluminescence mainly evolved as an anti-predation system [34,72,73,74,75,76,77]. However, in the light of recent data on unicellular photoreception, we argue that many aspects of bacterial and dinoflagellate bioluminescence point to additional functions. 

### 1.2. Perception and Responses to Light in the Unicellular World

The light-induced behavioural responses of microorganisms and fungi are mediated by a great variety of photoreceptors associated with diverse functions with gradual complexities [5,6,11,12,78,79,80,81,82,83,84,85,86,87]. While non-directional photoreception monitors variations in the light intensity, directional photoreception and low-resolution vision are used for orientation and habitat selection. High resolution vision constitutes the most elaborate system that enables complex behaviours such as prey detection and hunting [78,79,88]. In these behavioural responses to light, the photon activation of the light harvesting pigment is converted to an electric or chemical signal that is transmitted to the motor apparatus of the cells through modular cell signalling pathways such as the two component systems in bacteria [89,90,91,92] or phosphorylation cascades in eukaryotes [93,94]. Various systems have convergently evolved to provide accurate light-sensing such as the cellular micro-lenses of *Synechocystis* cyanobacteria used as micro-optics to sense light direction [95] or the elaborate eye-like ocelloids of certain dinoflagellates [96]. The best-characterised photoreceptors belong to different protein families and are associated with distinct chromophores [97] (Figure 1b). Despite their diversity, they share a similar property, which converts the ultrafast and local photo-activated changes in their pigment structure into long-lived changes in the receptor protein that are transmitted to other proteins in the signalling chain of the organisms [6,11]. The rhodopsin and photoactive yellow protein (PYP) use the photoisomerization of a C=C bond in retinal or the *p*-coumaric acid, respectively. Phytochromes use bilin, an open tetrapyrrole and the LOV proteins, cryptochromes and BLUF proteins use flavin mononucleotides (FMN) as a chromophore [98].

#### 1.2.1. The Great Diversity of Bacterial and Eukaryotic Photoreceptors

Rhodopsin consisting of seven trans-membrane α-helices belongs to the vast family of G-protein-coupled receptors [99,100,101,102,103,104,105]. The light inducing isomerization of the 11-cis-retinal is coupled to a cascade of events that ultimately acts as a switch to relay the signal to G-proteins [106,107], guanylate kinases [101,108] or phosphodiesterase [102] that triggers various intracellular responses. Sensory rhodopsins that are responsible for light signalling and photo-behaviours have been found in archaea and bacteria [109,110,111,112].

The PYP adopt a Per-ARNT-Sim (PAS) domain [113,114,115,116,117,118], a protein sensor superfamily for a vast range of stimuli, from light sensing to ligand binding [119]. The *trans* to *cis* photoisomerization of its *p*-coumaric acid chromophore proceeds with a “volume-conserved” mechanism that has been followed by time-resolved crystallography [120,121]. This primary-event induces small structural changes and is associated with small structural rearrangements associated with charge delocalization around the chromophore [122]. 

LOV (Light, Oxygen or Voltage) domains are photosensors that also adopt the PAS domain structure and use the FMN as a chromophore in bacteria, archaea and certain eukaryotes (plants and fungi) [7,80,123,124,125,126,127,128,129,130]. They can be found associated with various effectors in bacteria such as histidine kinases, proteins involved in the synthesis of cyclic guanosine monophosphate (cGMP), STAS (Sulfate Transporter and Anti-Sigma factor antagonist) and helix-turn-helix and have been found to be associated with other sensor domains [131]. Several signalling mechanisms have been proposed based on their dark and light-activated structures such as helix Jα unwinding [132,133] or dimerization [134].

BLUF, bacterial and eukaryotic Blue Light receptor Using Flavine adenine dinucleotide (FAD) have a modular architecture comprising a 150 amino acids (aas) receptor domain with a ferredoxin-like fold that can be connected with many different effectors whose activity is modulated by the BLUF domain in response to light [135,136,137,138,139]. The BLUF domains contains a flavin adenine dinucleotide chromophore. They can be found as single photosensory domains involved in light-regulated protein interactions [140] or in multidomain proteins fused to light-regulated transcriptional effectors [141,142], phosphodiesterases [135] or adenylyl cyclases [143,144]. The BLUF domain binds to the FAD/FMN/RF pigment non-covalently for the properties of the isoalloxazine to absorb blue light. In BLUF, a photo-induced proton-coupled electron transfer (PCET) initiates a rearrangement of the hydrogen bonds around the flavin cofactor after illumination that is transmitted to the surface of the receptor and leads to the activation of the various effectors.

Cryptochromes adopt a Rossman-like fold similar to photolyases [145,146,147,148,149,150,151,152,153,154,155,156,157,158,159,160] involved in functions ranging from DNA repair to the blue-light regulation of growth, development and circadian rhythms [161]. Cryptochrome photoreception is based on blue light-induced interconversion between several redox states of flavin as a chromophore [162]. While the role of tryptophan triad is controversial [163], the light-induced oligomerisation of the plant cryptochrome is an accepted mechanism for light-regulation interactions with various signalling partners [164,165,166].

Phytochromes are modular multi-domain red/far-red photosensory proteins that share conserved PAS, GAF and PHY core photosensory domains that adopt a knotted structure [167] and bind covalently to a linear tetrapyrrole chromophore (bilin, phytochromobilin or biliverdin) in a broad range of organisms such as bacteria, protists and plants [168,169,170,171]. Photoconversion triggers structural changes in the dimer interactions and the refolding of a “tongue” loop modulates the activity of the C-terminal “output” domains [172,173].

Light-sensitive ion channels involved in phototaxis by depolarizing the plasma membrane have been found in some eukaryotic unicellular algae (chlorophytes) [174,175]. These channels, such as the rhodopsin channel (737 aas), evolved from bacterio-rhodopsin consisting of seven transmembrane segments covalently linked to a retinal chromophore [176]. Light absorption induces the isomerization of the retinal chromophore, which by turning causes a set of conformational changes leading to the opening of a pore that allows ions to pass through [177,178,179]. In Cryptophyte algae, there are also cation-conducting rhodopsin channels that specifically conduct anions [180,181]; this group of channels is more structurally related to haloarchaeal rhodopsins and has different functional properties [182]. Light-sensitive channels have also been found in nucleocytoplasmic large viruses that infect plankton [183].

However, it has recently been shown that other types of receptors such as the chemoreceptors Aer, Tar, Tsr, Tap and Trg also mediate bacterial responses to blue-light [184]. In addition, a new photosensor family uses coenzyme B_12_ for sensing blue and green-light and regulating gene expression [185,186,187,188,189,190,191,192]. A bacteriochlorophyll usually involved in photosynthesis with bacteria has been found as a photosensor in the retina of dragon-fish *Malacosteus niger* living in the deep-sea [193]. Moreover, ChrR regulators mediate blue-light responses through the intermediate stage of the production of ROS in *V. cholera* [194]. These recent findings have considerably extended the landscape of photoreception in the microbial world. Interestingly, some photoreceptors and luciferase share the same pigments and therefore their metabolic pathways. For example, the flavin and bilin chromophores are found in the active sites of luciferases of some bacteria and dinoflagellates to catalyse the photon emission. Conversely, as found in many bioluminescent organisms, the pigment of photoreceptors may be not synthetised by the organism but is acquired from the external environment [195]. To date, the ultimate stage of unicellular high-resolution vision is achieved by eye-spots or ocelloids organelles observed in some dinoflagellate or *Chlamydomonas* species [96,155,196,197,198,199,200].

#### 1.2.2. Light and Unicellular Behaviours

Since the first observation of phototactic bacteria by Engelmann in 1883 [201], many studies have shown that light influences both the phototrophic and heterotrophic bacterial behaviours and physiology [8,9,11,202,203]. Phototaxis, the positive or negative displacement along a light gradient, displays various degrees of light-sensitivity among unicellular organisms including bacteria, archaea and eukaryotes [10,12,79,88,204,205,206]. Thus, very soon in the evolution, complex systems have emerged to accurately perceive and respond to light. Some phototactic archaea and cyanobacteria use photosensory rhodopsins to sense the intensity of light [112,207]. However, it has recently been shown that cyanobacteria represent the smallest and oldest organisms to use a kind of camera eye micro-optic for sensing light direction: individual *Synechocystis* cells directly sense the position of the light source through their own cells that act as micro-lenses [95,208] through cyanobacteriochromes [209,210]. Light can also induce collective behaviours in photosynthetic bacteria such as *Rhodospirillum centenum* [211] and other cyanobacteria [212,213].

In addition to phototaxis, light induces various physiological and behavioural responses in both photo and heterotrophic bacteria through their diverse photoreceptors [214]. This includes, for example, the light modulation of virulence, cell attachment, cell growth or biofilm formation [215,216,217,218,219]. Light plays an important role in modulating the bacterial behaviours and host–symbiont relationships in the rhizosphere [220]. Interestingly, the important effect of blue-light on bacteria has led to light-based anti-infective strategies being conceived [221,222,223,224]. However, particularly relevant for the present review is the finding that deep-sea bacteria that usually live in the non-photic zone possess photoreceptors and respond to light [225,226]. While it has been proposed that these deep-sea chemoreceptors may help to switch the behaviours between a planktonic and a benthic lifestyle, this finding may also support the hypothesis that bacteria perceive the bioluminescence of other deep-sea organisms, including themselves, which is the unique source of light in deep-sea sediments.

Light also modulates the phototaxis and circadian rhythms of multiple eukaryotic algae [83,161,227,228] but also induces various behavioural responses from other, non-photosynthetic protists such as dinoflagellates [198,229,230,231,232,233,234,235,236], ciliates [237,238,239]; affects biofilm formation by *Chlorella vulgaris* [240]; controls the morphogenesis in green plants and stramenopiles [241,242] and the sexual cycle in *Chlamydomonas* [243]; and influences the phototactic swimming in sponge larvae [244]. Interestingly, light also mediates cross-kingdom communication: it has recently been demonstrated that the green fluorescence of the cnidarian host attracts symbiotic algae [245] (Table 1)**.**

### 1.3. Communication, Collective Behaviours and “Cognition” in the Unicellular World

In his pioneer work “the psychic life of microorganisms” published in 1889 [246], Alfred Binet described the great behavioural richness of unicellular organisms that he observed under the microscope. He listed all the organs of locomotion and the sense organs that can be observed in protists and noticed their analogies with the corresponding organs in metazoans. He carefully described the movements, the complex and the species-specific behaviours of microorganisms and noticed, as Engelmann did in 1883 [201], the phototactic properties and sensitivity of bacteria to oxygen tension. A few decades later, in the United States, Jennings and Loeb also looked at and noticed the richness of the behaviour of what they called the lower organisms [247,248]. These works led to the realisation, which lasted for more than a century, that microorganisms, although lacking a nervous system, were capable of performing complex individual and collective behaviours. Today’s knowledge of the behavioural landscape of microorganisms goes far beyond what Binet could have imagined: microorganisms display sensitivities to diverse stimuli, they communicate with each other, they can change their behaviour according to their individual or collective way of life and are able to accomplish complex tasks such as associative learning and finding their way into a maze. The detailed knowledge of molecular sensors and signalling circuits now makes it possible to decipher how cells perceive, integrate and react to external and internal signals.

#### 1.3.1. Bacterial Communication and “Cognition”

Firstly, many studies have shown that unicellular organisms communicate with each other using a wide variety of chemical or physical signals. The best known system is quorum sensing that allows bacteria and other microorganisms to inform each other about their density and trigger and synchronise collective behaviour [249,250,251,252,253,254]. However, while chemical communication has been the first to be characterised, intra- and inter-cellular communication can also be established in a physical way by electrical, electromagnetic and acoustic waves or mechanical contacts [255,256,257,258,259,260,261,262,263,264,265,266]. For example, a fascinating study showed that biofilms formed by *Bacillus subtilis* could attract and incorporate motile cells of different species through the ion channel-mediated electrical signalling [259]. Other studies have provided evidence that voltage-gated signalling mediates long-range communication between bacteria [260,267].

Secondly, studies on bacterial chemotaxis have led to the realisation that bacteria can make decisions based on the integration of multiple signals and led the pioneers of chemotaxis studies to propose an analogy between these phenomena and neurobiology in the early 1980s [268,269]. Chemotaxis is the ability to choose its orientation according to a source of attractive or repulsive substances [270]. It integrates the perception of external or internal signals through two-component system signalling [89]. In these systems, a “sensor” protein auto-phosphorylates in response to a specific signal coming from specific receptors and transfers the phosphoryl group to a “response regulator” protein that carries out a cellular response [271,272,273,274,275,276]. These studies soon revealed “cognitive” properties of bacteria, showing that they could indeed make decisions in complex situations [270,277], amplify signals, show habituation faculties and retain the memory of past situations [278,279]. Later, on the basis of genomic studies, Michael Galperin demonstrated that bacteria display distinct behavioural types according to their genomes and ecosystems [271,280]. Today, many articles converge towards the universality of the notion of “consciousness”, “cognition” or “intelligence” in microorganisms [13,15,16,17,18,281,282,283,284,285,286].

#### 1.3.2. Complex Behaviours in Unicellular Eukaryotes

More elaborate behaviours that are close to that of metazoans have been observed in unicellular eukaryotes. Habituation, the simplest form of learning that consists of attenuating the response and eventually ceasing to respond to a repeated stimulus that has proven not to be harmful was initially characterised in a mollusc, the aplysia [287]. Habituation has been recently observed in organisms lacking a nervous system [288]. More elaborate, associative learning establishes a link between the joint appearances of different stimuli and associates them. This faculty is the basis of what is known as Pavlovian conditioning. In the middle of the 20th century, Beatrice Gelber demonstrated that, as with Pavlov’s dogs, *Paramecia* could be conditioned to link two different stimuli [21]. Although initially contested, these fascinating studies have recently been rehabilitated [22]. Since these pioneering studies, growing evidence has shown that they are not exclusive to beings with a nervous system and that complex behaviours such as associative learning and the solving of complex tasks are observed in unicellular organisms [19,20,22,289,290,291,292].

## 2. Hypothesis: Bioluminescence Signalling in the Unicellular World

Whereas chemical communication has long dominated the literature about signalling between single-cell organisms, many studies have recently shown that cells can also exchange physical signals such as sounds, electrons or electromagnetic waves [255,256,257,258,264,266,293,294]. However, while the influence of ambient light on the behaviours of microorganisms is now well established, communication by light emitted by luciferases of bioluminescent unicellular organisms has not been previously addressed. Previous studies have brought experimental evidence that non-bioluminescent prokaryotic and eukaryotic unicellular organisms such *E. coli* and *Paramecia* communicate between themselves through the emission and reception of weak photons [264,295,296,297]. It is therefore likely that organisms that have evolved specific bioluminescent systems to emit more intense light and modulate light intensity communicate with each other or with other species through light. Thus, bioluminescence could also contribute to survival strategies involving light signalling within the unicellular world. Bioluminescence produced by bacterial or eukaryotic unicellular organisms is therefore expected to produce various kinds of intra- or inter-species behavioural responses in cells that possess photoreceptors or even more elaborate high-resolution vision such as eye spots or ocelloids [96,196,197]. Thus, we propose here that the interplay between bioluminescence and photoreception should be systematically explored from both an evolutionary and functional perspective such as, for example, in light-signalling within a single or among different species and taxa. Of course, bioluminescence is expected to exert an effect on other organisms in a particular ecological context protected from ambient-light: in the non-photic zones of the ocean, during the night, into the internal organs of animals or in the plant rhizosphere (Figure 2).

### 2.1. Light Signalling in Bioluminescent Bacteria

Most of the bioluminescent bacterial species fall within the Gammaproteobacteria class and cluster phylogenetically in three families (*Vibrionaceae, Shewanellaceae* and *Enterobacteriaceae*) which all carry the canonical highly conserved *lux* operon [32,62]. Since its first identification more than 40 years ago, it has been shown that the *lux* operon adopts a “canonical” *luxCDAB(F)E(G)* organization that has been systematically observed in genomes of bioluminescent bacterial. The *luxA* and *luxB* genes encode the α and β luciferase subunits that emit light by the oxidation of FMNH_2_ and a long chain aldehyde [38,61]. LuxC, D and E proteins form a fatty acid reductase complex that synthesises the long chain aldehyde substrate. The recent discovery of new *lux* operon organisations in other bacterial taxa has opened new evolutionary and functional perspectives on bacterial bioluminescence [60].

Bioluminescent bacteria occupy a large variety of habitats and lifestyles [62]. It has long been thought that the genes involved in bioluminescence catalysis and control have co-evolved to adapt to various ecological niches [298,299,300]. However, many aspects of the evolution and functions of bacterial luminescence remain enigmatic. It is currently thought that the bacterial emission of light is essentially intended for the visual systems of their infected or symbiotic hosts (see above). However, recurrent features indicate that light plays a critical role in host–symbiont communication and in the formation of bacterial biofilms on host tissues by animal photoreception systems other than visual systems. For example, *Vibrio fischeri* establishes a complex symbiotic relationship with the squid *Euprymna scolopes* in which the bacterial luminescence is perceived by photoreceptors within the light organ of the host [70,301]. Thus, *V. fischeri* cells defective in light production are not retained for symbiosis and are expelled from the organ [302,303]. In addition, the control of bioluminescence by the regulation of the *lux* operon has also revealed some surprises. While it was initially thought that the *lux* operon is dependent on cell density through quorum sensing [251], it has recently been found that in some species, light-emission is independent of quorum sensing [304] or controlled by other factors depending on the environmental conditions [305,306,307,308,309,310]. 

In *Photorhabdus luminescens*, the only terrestrial luminescent bacteria, light-emission is specifically associated with a specific symbiotic stage of its life cycle. *P. luminescens* switches between different phenotypic traits associated with a symbiotic life in the gut of the nematode *Heterorhabditis* (primary 1° cells), the infection of insect larvae (2° cells) and the plant root in the rhizosphere [311,312,313]. Only the primary cells (1°) that correspond to cell clumping and the symbiotic stage with the nematode is bioluminescent suggesting that light emission is required in communication with its host. However, little is known about its bioluminescence during the rhizosphere life stage, since this life form has only recently been discovered [313]. Knowing that many soil bacteria display photoreceptors, it is possible that the bioluminescence of *P. luminescens* plays a role in its association with the rhizosphere bacterial community. In addition, it has recently been found that other bacterial species possess non-canonical *lux* operons and are probably luminescent [60]. Some of these species are also frequently associated with the rhizosphere, thus reinforcing the idea that light may play an important role in cell-to-cell communication among the different species of the rhizosphere. Interestingly, several studies have shown that plant roots also possess multiple photoreceptors that sense different wavelengths including blue-light and display various behavioural and physiological responses to light [314]. This suggests that the bioluminescence of the bacteria that live in the rhizosphere may not only influence other root associated bacteria but also affect the growth and behaviour of root apices (Figure 3). Furthermore, the existence of several species of bioluminescent earthworms [315,316,317] reinforces the idea that the emission and perception of light in soils could play an ecological role that would be interesting to elucidate in more detail.

In addition, the biofilm formation of bioluminescent bacteria isolated from the marine fish gut has recently been described [318]. This study shows that the bioluminescent bacteria *Kosakonia cowanii* forms cross-species biofilms with Gram-Positive coccoid bacteria. Does light emission play a role in this association in the gut? An important hint that supports our hypotheses is that bacteria from the deep-sea, and therefore living in the non-photic zone, possess photoreceptors that mediate light-behavioural responses. For example, it has been shown that some deep-sea species possess either PYP or bacteriophytochrome photoreceptors that regulate biofilm formation or growth [225,226]. While these studies have proposed that these photoreceptors contribute to the regulation of their benthic and pelagic lifestyle, they may also make bacteria sensitive to bioluminescence in the non-photic zone. This suggests that bioluminescence in the deep-sea could play a signalling role in the formation of biofilms or modulate the behaviours of other benthic bacterial species.

Since photons can be perceived by the multiple photoreceptors found throughout the tree kingdoms of life, bioluminescent bacteria may also have evolved a selective advantage through light-signalling with both prokaryotic and eukaryotic unicellular organisms. They could for example repel photophobic predators such as certain ciliates [239] but they could also contribute to the selection of the bacterial consortium around them: light may attract photophilic or repel photophobic species (Figure 3). Light emission and perception may therefore contribute to the shaping of bacterial communities and their symbiotic associations with various eukaryotic organisms. Thus, light signalling may be a general property of the bacterial collective way of life and could mediate their symbiotic associations with animal, fungi or plants [71,214,319]. Light communication could for example contribute to the formation of algae/bacteria blooms [320] such as the spectacular milky sea, whose bioluminescence can be observed from space [321,322] or bioluminescent blooms that occur in the deep-sea [323]. Thus, following on from Seliger’s and O’Kane’s works [299,300], it would also be interesting to study the co-evolution of genes involved in bioluminescence and photoreception in bacteria, or even the co-evolution of these systems between symbiotic organisms.

### 2.2. Do Dinoflagellates Converse by Light Signals?

Bioluminescent dinoflagellates, which are the major contributor to stimulated light-emission in the upper ocean, are currently the best-characterised luminescent unicellular eukaryotes, both on a molecular and ecological scale [33,34,74]. Among the 1555 species of dinoflagellates, there are ~70 bioluminescent species, including both autotrophic and heterotrophic species, occupying diverse oceanic niches [33,34,324]. The bioluminescence of dinoflagellates is part of numerous excitable responses to a variety of physical or chemical stimuli leading to complex individual or collective behaviours. Light emission occurs in specific organelles, the scintillons. They consist of dense vesicles of about 0.5–0.9 μm in diameter that contain the luciferase enzyme (LCF), the luciferin substrate and in some species, a luciferin binding protein (LBP) [325,326,327]. Mechanical stress on the cell membrane triggers an action potential on the vacuole membrane of their associated scintillons through a mechanotransduction pathway involving G-proteins and calcium signalling [328,329,330,331]. The action potential opens the voltage-gated proton channels and induces a rapid and transient acidification of the scintillons [332] where the catalytic oxidation of luciferin, a chlorophyll-derived open tetrapyrrole, by pH-regulated luciferases produces light flashes [40]. The emitted light has a wavelength in the range of 474–476 nm with a wide range of duration and intensities depending on the species. While the large dinoflagellate species such as *P. noctiluca* (~ 350 μm) or *P. fusiformis* (up to 1000 μm) can emit long (200–500 ms) and bright flashes (from 3.7 × 10^10^ to 6.5 × 10^14^ photon cell^−1^), smaller cells such as *L. polyedra* (~20–40 μm) emit brief (~100 ms) and dim flashes (from 3.1 × 10^7^ to 1.2 × 10^8^ photons cell^−1^) [324,333,334]. From membrane deformation to light emission, the overall bioluminescent response is rapid and can take only 12 to 20 μs [72,328]. The bioluminescence response threshold or flow sensitivity also depends on the size and morphological features of the different species [335]. 

Although these phenomena are well described from a mechanistic point of view, there are still many points to be clarified, in particular with regard to the ecological function of dinoflagellate bioluminescence. Overall, the current paradigm is that dinoflagellate bioluminescence has evolved as an anti-predation system. Three major hypotheses that are not exclusive have been considered and tested experimentally (reviewed in [33,34,58,324]): (i) the startle response, (ii) the burglar alarm and the (ii) aposematic warning. The first hypothesis proposes that the emission of light protects prey by frightening the predators. In the “burglar alarm” hypothesis, it is thought that the bioluminescence emitted in the presence of an herbivore (predator of the dinoflagellates) provides a selective advantage by attracting and allowing a carnivore (predator of the herbivore) to perceive to capture the herbivore, to the benefit of the dinoflagellate population. In the aposematic warning, light is used as a “grazing deterrent”, a lure that signals to predators that the prey may be dangerous, toxic or has unpalatable traits. While several experimental studies support some of these hypotheses [75,333,336] some characteristics of dinoflagellate bioluminescence may suggest that it could also have other functions. For example, some strains of *Noctiluca scintillans* off the west coast of the USA have lost their bioluminescence but are still able to form dense populations despite the presence of predators, thus questioning the ecological role of bioluminescence in this species [337]. Moreover, some studies have reported that many species of dinoflagellates can spontaneously emit light flashes, without being mechanically stimulated [334,338,339]. Furthermore, 30 years ago, three studies indicated that artificial light flashes on dinoflagellate cultures could induce their bioluminescence [340,341,342] (Table 1). These observations therefore call for a closer look at the inter-relationships between the ability of dinoflagellates to emit and perceive light and their possible functional consequences.

#### 2.2.1. Photoreception and Light-Induced Behaviours in Dinoflagellates

In addition to their bioluminescent capabilities, dinoflagellates can perceive and distinguish between different light wavelengths and can exhibit a number of behavioural and physiological responses to light exposure (Table 1). Circadian rhythms [161,343], whose clock is generally light-regulated, control many functions such as bioluminescence, cell-cycle, collective behaviours and daily vertical migration along the water column. For example, bioluminescence is generally inhibited during the day and its intensity during the dark phase may depend on the amount of illumination during the previous day. However, a distinct phenomenon, the “photoinhibition” of bioluminescence has also been observed: this consists of the transient suppression of stimulated bioluminescence when cells are exposed to intense light [344,345,346]. Conversely, some studies have shown that artificial laser flashes may induce the light-emission of several dinoflagellate species such as *P. lunula*, *P. fusiformis* and *G. polyedra* [340,341,342]. Many species also display positive or negative phototactic behaviours at different wavelengths [12,230,232] (Table 1). The influence of light on the mobility of symbiotic species such as *Symbiodinium sp.* also plays an important role in symbiotic relationships with corals. It has been shown, for example, that some *Symbiodinium* species are specifically attracted to the colours emitted by the fluorescent proteins of their coral hosts. In contrast, a stop-response, the abrupt cessation of movement induced by intense light was also observed in *Gyrodinium dorsum* [347,348].

**Table 1 ijms-22-11311-t001:** Influence of light on dinoflagellate behaviours.

Title	Light Response	Species	References
**Diurnal rhythms**			
	Bioluminescence is under circadian control	*Gonyaulax polyedra*	Sweeney and Hasting, 1957 [349] Hasting and Sweeney, 1958 [350]
			Roenneberg and Hasting, 1988 [351]
			Roenneberg and Deng, 1997 [352]
			Roenneberg and Taylor, 1994 [353]
			Morse et al., 1989 [354]
			Krasnov et al. 1980 [339]
		*Pyrodinium bahamense* *Gonyaulax polyedra* *Pyrocystis lunula*	Biggley et al. 1969 [334]
		*Mixed dinoflagellate communities:* *Gonyaulax spp.* *Alexandrium spp.* *Ceratium fusus* *Pyrocystis spp.* *Protoperidinium spp.*	Marcinko et al., 2013 [355]
		*Multiple bioluminescent species*	Backus et al. 1961 [356] Yentsh et al., 1964 [357]
	Photoenhancement of bioluminescence	*Ceratium fusus*	Sullivan and Swift, 1995 [236]
	Photoinhibition of bioluminescence	*Alexandrium catenella*, *A. acatenella*, *A. tamerensis*	Esaias et al., 1973 [344]
		*Gonyaulax polyedra*	Hamman and Seliger, 1982 [345]
		*Ceratius fusus*	Sullivan and Swift, 1994 [346]
		*Protoperidinium spp.*	Buskey et al., 1992 [358]
**Laser-induced bioluminescence**			
	Artificial red flashes induce bioluminescence (wavelength 585 nm)	*Pyrocystis lunula*	Hickman and Lynch, 1981; Hickman et al. 1982 [340,341]
	Detailed study of flash-induced bioluminescence	*Pyrocystis lunula*, *Pyrocystis fusiformis*, *G. polyedra*	Sweeney et al., 1983 [342]
**Gene expression induced by light**			
	Xanthorhodopsin subgroup II: proton pump for energy supplement during light-limited photosynthesis	*Prorocentrum donghaiense*	Shi et al., 2015 [359]
		*Oxyrrhis marina *	Guo et al., 2014 [360]
	A few genes are only transcriptionally regulated by light	*Symbiodinium kawagutii*	Zaheri et al., 2019 [361]
	Light-transcriptional control of 9.8% of the genes	*Karenia brevis*	Van Dolah et al., 2007 [362]
	Different photoresponses according to species	*Dissodinium lunula*, *Pyrocystis fusiformis*, *Pyrocysti noctiluca*	Swift and Meunier, 1976 [363]
		*Gonyaulax tamarensis* *Heterocapsa trique*	Anderson and Stolzenbach, 1985 [364]
**Motility, phototaxis**			
	Swarming, diel vertical migration	*Gonyaulax polyedra*	Roenneberg et al., 1989 [365]
	Light induces “stop-responses” or “shock reaction” (cessation of movement)	*Girodinium dorsum*	Forward and Davenport, 1968; 1970 [348,366]
	Stop-response followed by positive phototaxis	*Gymnodinium splendens*	Forward, 1974 [233] Ekelund and Björn, 1987 [347]
	Positive-Phototaxis (Comparison of phototaxis in dinoflagellate species with and without eyespots)	*Scrippsieiia hexapraecinguia* *Peridlnium foliaceum* *Atexandrium hiranoi* *Gymnodinium mikimotoi*	Horiguchi et al. 1999 [367]
	Photoresponse of an heterotrophic dinoflagellate in three wavelengths: 450 nm, 525 nm and 680 nm) mediated by rhodopsin	*Oxyrrhis marina*	Hartz et al., 2011 [231]
	Modulation of phototactic and stop response by wavelengths	*Girodinium dorsum*	Hand et al., 1967 [368]
	Support the hypothesis of a two-pigment system in phototactic response	*Girodinium dorsum*	Forward, 1973 [232]
		*Peridinium gatunense*	Häder et al., 1990 [234]
**Effect of light on symbiosic species**			
	Blue-light has a dominant effect on the cell cycle	*Symbiodinium spp.*	Wang et al., 2008 [369]
	Physiological adaptation to light gradient in corals	*Symbiodinium spp.*	Wangpraseurt et al., 2016 [370]
	Green light facilitates symbiont capture by coral larvae	*Symbiodinium spp.*	Hollingsworth et al., 2005 [371]
	Symbiodinium species are specifically attracted by the green fluorescence emitted by its coral host	*Symbiodinium spp.*	Aihara et al., 2019 [245]
	Variable phototaxis responses according to different *Symbiodinium* species	*Symbiodinium spp.*	Yamashita, 2021 [372]
	Optical feedback loop involving dinoflagellates and coral in coral bleaching	Multiple species	Bollati et al., 2020 [373]
	Eye-spot and spectral sensitivity of phototaxis	*Kryptoperidinium foliaceum*	Moldrup and Garl, 2012 [230]

How do dinoflagellates perceive light? Similar to most of the planktonic organisms [78,374,375], dinoflagellates possess various kinds of photoreceptors and some species possess elaborate vision systems such as eye-spots or ocelloids [96,199,376]. Identified in many genomes of distantly related species, rhodopsin is considered ubiquitous in dinoflagellates [377,378,379,380,381,382]. For example, in the genome of *O. marina*, two forms of rhodopsins have been identified: proteorhodopsins related to bacteriorhodopsins [107,383,384], which probably function as proton pumps and sensory rhodopsins [360,385]. It has been proposed that proton pump proteorhodopsins contribute to energy resources and compensate photosynthesis in light deprived environments [105,359,386]. On the other hand, sensory rhodopsins [111,112] are more likely to have a function in photoperception and signalling. Interestingly, rhodopsins have been also found in the retinal body of *Erythropsidinium* eye-spots [376]. Knowing that rhodopsins found in the eye-spot membrane of *Chlamydomonas* and *Volvox* mediate phototaxis [387] or complex light-behavioural responses in *Peranema* [235], it is likely that they play a similar role in dinoflagellates. Supporting this idea, it has been shown that the outer cell membrane rhodopsins of the heterotrophic dinoflagellate *O. marina,* mediate the photosensory response to detect algal prey based on chlorophyll autofluorescence [231].

A cryptochrome blue-light receptor belonging to the CRY-DASH family has also been identified in *Karena brevis* [388]. While cryptochromes are generally known to be involved in circadian control [155,156], this family of photoreceptors may also play a role in the phototactic behaviour of sponges [244]. This opens interesting perspectives on other potential functions of cryptochromes in dinoflagellates.

Although phytochromes are generally absent from dinoflagellates, transcriptomic study has also found a phytochrome gene in the diatom symbiont of the dinotom *Durinskia baltica* [169,389,390]. The most spectacular vision systems observed in unicellular organisms are the eye-spots and ocelloids that have evolved in *Erythropsidinium* and *Warnowia* [96,376,391]. They have subcellular analogues to cornea, lens, iris and retina similar to vertebrate eyes that evolved from mitochondria and plastids. They are thought to provide a cellular high-resolution vision and allow the cells to react to the intensity and direction of light.

#### 2.2.2. Excitability, Stimuli Integration and Complex Behaviours in Dinoflagellates

In addition to acute light-perception, growing evidence suggests that eukaryotic unicellular organisms including planktonic ones are able to integrate multiple chemical or physical stimuli to produce appropriate behavioural responses in their fluctuating environments. Many studies have reached the conclusion that the “behaviour evolved before nervous systems” [392] and the “neural aspects of biological systems were present in single-celled organisms” [393]. Interestingly, both bacteria and protists are excitable cells that perceive and propagate stimuli through voltage-gated ion channels similar to those of the animal nervous systems [267,394]. For example, a recent study showed that the diverse families of dinoflagellate H_v_1 proton channels that were initially thought to control bioluminescence may also play diverse cellular roles [395]. Molecular networks in the cells have probably evolved to integrate the multiple stimuli and mediate complex behavioural responses such habituation, associative learning or decision-making, abilities suitable for the establishment of elaborate modes of communication [26,27,28,29,396]. 

Indeed, dinoflagellates exhibit a rich real-time behavioural and morphological repertoire to adapt to their diverse and fluctuating environmental conditions [397,398]. For example, in addition to the high-resolution vision systems described above [96,376], some dinoflagellate species have developed complex organs such as a piston for efficient propulsion [198] or nematocysts for hunting [399] that require complex “sensorimotor” and coordination molecular processes. In bioluminescent dinoflagellates, excitable light emission is stimulated by mechanical forces that trigger an “action potential” and a subsequent signalling cascade leading to the activation of luciferase in the scintillon [34,72,328,331,332,400]. Such excitable responses share both mechanistic and functional analogies with that of animals that possess nervous systems [401,402,403] and interestingly, the dinoflagellate responses are modulated by the neuromediators or inhibitors of voltage gated channels found in animal nervous systems [404,405]. In addition, it has been shown that some dinoflagellate species change their rheotactic behaviours, actively diversify their swimming strategies, form long multicellular chains or have the capacity to adopt compensating morphological countermeasures in response to variations in hydrodynamic environments [397,406,407,408]. Dinoflagellates also display collective and concerted behaviours such as the transition between life cycle stages during blooms or the formation of temporary cysts for escaping parasite infections [409,410]. Heterotrophic species such as *Oxyrrhis marina* display complex feeding behaviours such as prey discrimination and preference [411,412]. These studies that reveal the elaborate cognitive capacities of dinoflagellates are conducive to the consideration of new functions for their bioluminescence. 

#### 2.2.3. Photosensing of Bioluminescence: Emitting Light for Signalling

Knowing firstly that dinoflagellates possess various photoreceptors or even sometimes highly developed vision systems, secondly that some species have developed various types of responses to light stimuli and thirdly, that they can demonstrate highly elaborate individual and collective behaviours in the face of fluctuating situations in their environment, it seems likely that dinoflagellates could perceive, integrate and respond to the light stimuli emitted by their own bioluminescence, at night or in the non-photic zone (Figure 2). Interestingly, a recent study has shown that the photoreceptors of eukaryotic plankton communities are expressed predominantly at night, at a time that coincides with the maximum spontaneous (not externally stimulated) bioluminescence [413]. This study therefore supports the idea that these photoreceptors are dedicated to the perception of the bioluminescence that is most often under circadian control and emitted during the night [355,414,415]. 

On the other hand, some clues in the known properties of light emission in different dinoflagellate species may support this hypothesis. While it is now well known that the intense flashes produced by dinoflagellates are triggered by mechanical stress or flows to startle the copepod predators [75,76,328,329,330,405,416], it cannot be excluded that the modulation of flash intensities or durations may play more complex roles in intra- or inter-species communication. Moreover, some early works have shown that mechanical stress is not the only cause of light emission in dinoflagellates. For instance, dinoflagellates can spontaneously emit flashes or weak glowing without any apparent stress [334,338,339]. While it is thought that these spontaneous light emissions are stochastic, one cannot exclude the possibility that some cells voluntarily trigger these phenomena for a reason that is still unknown. Other works have also shown that dinoflagellate light-emission is not simply an all or nothing phenomenon: the intensity of the flashes can vary according to whether the induced flow is turbulent or laminar [417,418,419] or can depend on amplitude and the rate of cell wall deformations [420]. It is therefore possible that the intense flashes induced by stress contain more nuanced types of light emissions playing other functional roles which have not yet been investigated. The modulation of flash duration and intensity therefore also points towards the informational potential to communicate through light. 

However, the emergence of communication during evolution is intimately linked to the concomitant development of a social life in various organisms. For example, communication through visual signals is widespread in the animals that display various degrees of social behaviours. Many properties of light such as colours, skin patterns, iridescence and bioluminescence have been developed by evolution for intra- or inter-species signalling [421,422,423,424,425,426,427,428]. While intra-species communication generally involves social interactions and mating, inter-species communication mediates symbiotic relationships or shapes ecosystem interactions. It turns out that many species of dinoflagellates have both a “social” and sexual life [429,430,431,432] that may require an animal-like “language”. One hypothesis could be, for example, that dinoflagellate bioluminescence displays some analogies with light-emission strategies used by various bioluminescent animal species to communicate [433,434,435,436,437,438,439,440,441]. Interestingly, complex collective behaviours involving communication for mating have been observed in other unicellular organisms such as ciliates [442,443,444]. It would therefore be interesting to investigate which types of light signals bioluminescent dinoflagellates emit “spontaneously” with modern technology using single-cell systems. In summary, dinoflagellates have the “cognitive abilities” and “social behaviour” that suggest that they could use light as a means of intra- and inter-species communication. Establishing a parallel between light emission in the macroscopic and microscopic world may bring some heuristic value for deciphering the still mysterious bioluminescence of unicellular planktonic organisms (Figure 4).

## 3. Conclusions

Whereas chemical communication has long dominated the literature about signalling between microorganisms, many studies have shown that cells are also sensitive to physical signals such as sounds, electrons or electromagnetic waves. For example, ambient light is known to induce various responses in microorganisms through their large variety of photoreceptors. However, how bioluminescence is perceived and influences unicellular communication and behaviours remains to be explored. The functions of unicellular bioluminescence organisms are still enigmatic and it is thought that light-emission is mainly directed at visual animals, either as an anti-predator strategy, as for example in dinoflagellates, or in establishing symbiotic relationships in bacteria. In this short review, the interplay between bioluminescence and the photoreception of microorganisms has been examined in the light of numerous studies demonstrating their behavioural richness. We argue here that many aspects of bacterial and dinoflagellate bioluminescence point to additional functions and propose that similar to visual animals, feed-back loops between light-emission and reception could play multiple roles in intra- or inter-specific communication and participate in complex behaviour in the unicellular world. In addition, bioluminescence and photoreception could play a general role in the communication between symbiotic microorganisms and their plant or animal hosts.

## Figures and Tables

**Figure 1 ijms-22-11311-f001:**
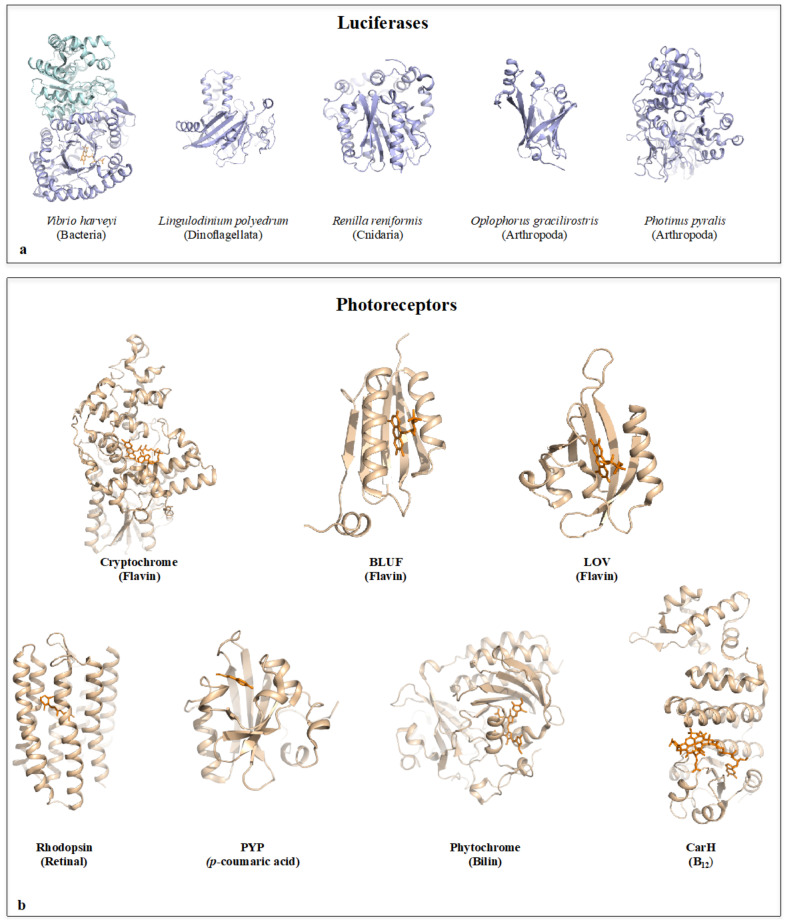
Structures of luciferases and photoreceptors. (**a**) the structures of the luciferases are depicted with blue cartoons. At the periphery, the various types of photoreceptors are represented with beige cartoons. The corresponding pdb codes for the luciferases are: *Vibrio harveyi*: 3fgc, *Lingulodinium polyedrum*: 1vpr; *Oplophorus gracilirostris*: 5b0u; *Photinus pyralis*: 1lci; *Renilla reniformis*: 2psh. (**b**) The structures of the main classes of photoreceptors are represented by wheat cartoons. The pdb codes for the photoreceptors are: Cryptochrome: 1np7; BLUF: 1yrx; LOV: 1g28; Phytochrome: 2o9c; CarH: 5c8f; PYP: 2phy; sensory rhodopsin: 1jgj. The co-factors and pigments are indicated and represented with orange sticks if present in the structures of both luciferases and photoreceptors. The pigments are written in parentheses.

**Figure 2 ijms-22-11311-f002:**
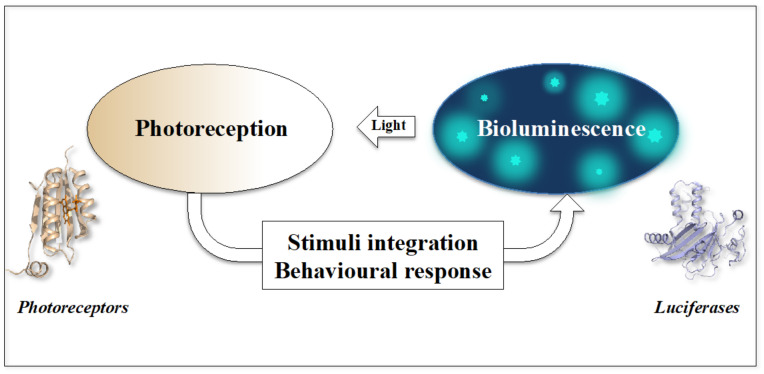
Interplay between photoreception and bioluminescence in microorganisms. Schematic diagram of the feed-back loop between light emission and light reception. During the night, in the non-photic zone or in the rhizosphere (soil), light emission by luciferases (symbolised here by dinoflagellate luciferase) and light reception by photoreceptors (symbolised here by BLUF) has the potential to mediate intra- or inter-species light-communication.

**Figure 3 ijms-22-11311-f003:**
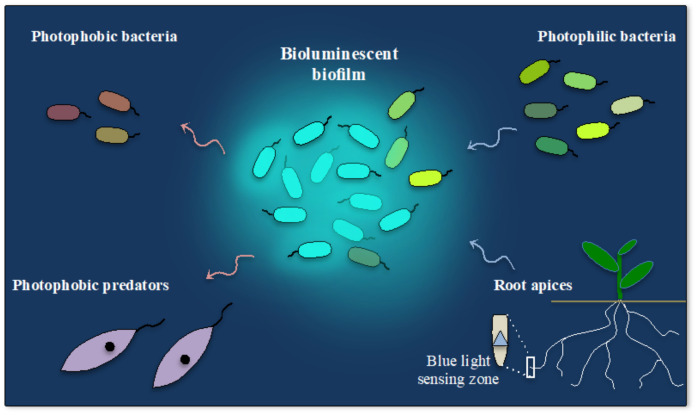
Light signalling in the bacterial world. Schematic representation of the hypothesis that bioluminescent bacteria may shape the bacterial communities in their luminous biofilm (represented by a cyan halo). They can attract different species of photophilic bacteria (green) or repell photophobic bacteria (brown) and eukaryotes (violet). Thin, wavy arrows symbolise cell motility. Light emission may also influence the plant roots that possess blue-light photoreceptors (blue triangle) (see [314]).

**Figure 4 ijms-22-11311-f004:**
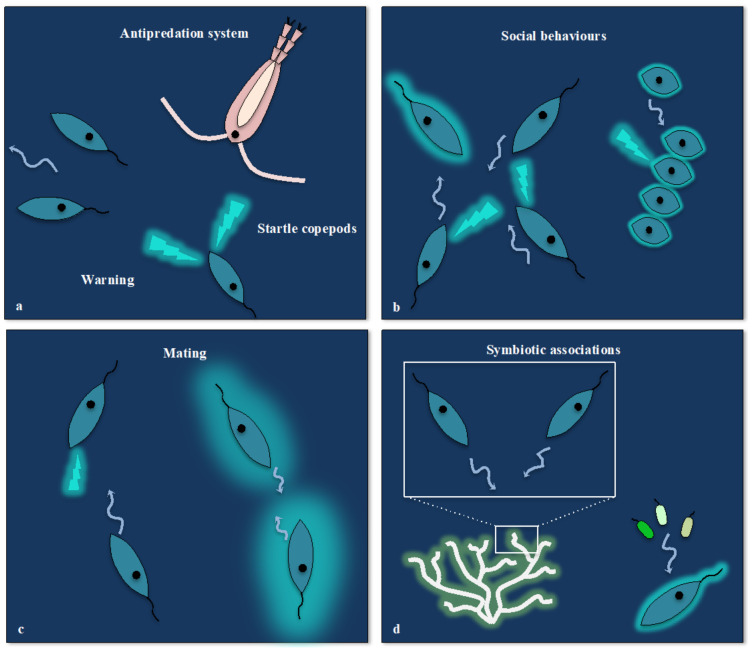
Schematic representations of the multiple potential functions of dinoflagellate bioluminescence and photoreception. (**a**) The intense emitted flashes play an anti-predation role such as startling copepod predators. They can also alert other cells to danger and provide information about the location and the nature of predators (warning). (**b**) Flashes and glowing (cyan halo) can contribute to cell signalling in complex social behaviours such as blooms (left) or chain formation (right). (**c**) Glowing and the modulation of flash intensities and durations may contribute to communication for mating. (**d**) Light can also mediate symbiotic associations with coral (left) or bacteria (right).

## Data Availability

Not applicable.

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
