# Peer review of "Bioluminescence and Photoreception in Unicellular Organisms: Light-Signalling in a Bio-Communication Perspective"

_ijms, 2021, doi:10.3390/ijms222111311_

Round 1
Reviewer 1 Report
The review paper by Timsit et al., entitled “Bioluminescence and photoreception in the unicellular world: light-signalling in a bio-communication perspective” interestingly describes bioluminescence and photoreception in bacteria and dinoflagellates from the viewpoint of biology. The focusing topic is significant to the field of bioluminescence and this review paper will be of interest to readers of this journal. However, this reviewer has the following concerns.
Major points:
1) On page 1, the sentence “Here, we focus our analysis on ... whose bioluminescence is well characterised" in the abstract. The word “characterised” should be changed to the word “studied” because the details of molecular mechanism of dinoflagellate bioluminescence remains to be clarified.
2) On page 2, the sentence “Light-emission is usually produced by ... such as bacteria, dinoflagellates, cnidarian, arthropods and fungi.” in the section 1.1. The photoprotein aequorin should not be treated as one of luciferases. In addition, cited references seem not suitable. At least, the following papers should also be cited. The papers: Nakamura et al., Structure of dinoflagellate luciferin and its enzymatic and nonenzymatic air-oxidation products, JACS 1989, 111, 7607-7611; Wu et al., Solution structure of Gaussia luciferase with five disulfide bonds and identification of a putative coelenterazine binding cavity by heteronuclear NMR, Sci. Rep. 2020, 10, 20069. Furthermore, throughout the Introduction section, the authors should carefully check whether cited references are suitable or not.
3) On page 2, in the second paragraph of the section 1.1. The authors should carefully mention luminous bacteria symbiosis because bioluminescence based on symbiotic bacteria is important as shown in the following review paper. The paper: Tanet et al., Reviews and sysntheses: Bacterial bioluminescence-ecology and impact in the biological carbon pump, Biogeosciences 2020, 17, 3757-3778.
4) On page 4, Figure 1 is confusing. Is the relationship between each luciferase and each photoreceptor clear? What does the color of arrows mean? Photoreceptors and luciferases might be illustrated separately in different panels.
5) On page 8, the sentence “light-emission and photoreception may have probably co-evolved in a signalling context...” in the first paragraph of the section 2. The word “co-evolved” seems too strong expression. The sentence should be rephrased.
6) On page 8, Figure 2 is confusing because light-emission and photoreception seem to be evolved under ambient-light condition. The authors should consider the illustration.
7) On page 9, the sentence “It is therefore possible that ...association with the rhizosphere community” in the third paragraph of the section 2.1. In soil, scattering of light seems not to occur. In addition, other luminous organisms (e.g. luminous earthworm) are in soil, and there is a possibility that light from such organisms affects the rhizosphere community or roots of plants. The authors should consider these points.
8) On page 10, Figure 3 is confusing because plants seem to indirectly affect photophobic predators. The authors should consider the illustration.
9) On page11-13, Table 1 is confusing and not well organized. For example, many empty fields are confusing. The authors should carefully correct the table.
Minor points:
1) In the title, the wording “in the unicellular world” seems impressive expression, but may be too strong expression. This reviewer thinks that the wording “in unicellular organisms” might be better.
2) On page 3, the sentence “The trans to cis photoisomerization of its p-coumaric acid...” in the second paragraph of the section 1.2.1. “p” of the word “p-coumaric acid” should be in Italic. In addition, throughout the paper, the authors should check typos.
3) On page 6, the sentence “this finding also supports the notion that bacteria perceive bioluminescence ... in deep-sea sediments” in the second paragraph of the section 1.2.2. The sentence maybe too strong. This reviewer thinks that the sentence should be toned down a little or rephrased.
Author Response
Above all, we would like to thank the reviewers for their valuable advice and suggestions to improve the manuscript. We have followed each recommendation regarding text, references and figures. Changes in the revised manuscript are indicated in red.
Reviewer 1
Major points:
1) On page 1, the sentence “Here, we focus our analysis on ... whose bioluminescence is well characterised" in the abstract. The word “characterised” should be changed to the word “studied” because the details of molecular mechanism of dinoflagellate bioluminescence remains to be clarified.
The word « characterised » has been changed by « studied » in the abstract of the revised version
2) On page 2, the sentence “Light-emission is usually produced by ... such as bacteria, dinoflagellates, cnidarian, arthropods and fungi.” in the section 1.1. The photoprotein aequorin should not be treated as one of luciferases. In addition, cited references seem not suitable. At least, the following papers should also be cited. The papers: Nakamura et al., Structure of dinoflagellate luciferin and its enzymatic and nonenzymatic air-oxidation products, JACS 1989, 111, 7607-7611; Wu et al., Solution structure of Gaussia luciferase with five disulfide bonds and identification of a putative coelenterazine binding cavity by heteronuclear NMR, Sci. Rep. 2020, 10, 20069. Furthermore, throughout the Introduction section, the authors should carefully check whether cited references are suitable or not.
The reference and text about aequorin have been suppressed and the proposed references on dinoflagellate luciferin (ref 41) and NMR structure of Gaussia luciferase (ref 48) have been added in the revised version. We have reorganised the text to cite more appropriately the references about bioluminescence. We have added a reference about the verterbrate bioluminescence (ref 37).
3) On page 2, in the second paragraph of the section 1.1. The authors should carefully mention luminous bacteria symbiosis because bioluminescence based on symbiotic bacteria is important as shown in the following review paper. The paper: Tanet et al., Reviews and sysntheses: Bacterial bioluminescence-ecology and impact in the biological carbon pump, Biogeosciences 2020, 17, 3757-3778.
The reference of Tanet et al. has been added (ref 69)
4) On page 4, Figure 1 is confusing. Is the relationship between each luciferase and each photoreceptor clear? What does the color of arrows mean? Photoreceptors and luciferases might be illustrated separately in different panels.
The figure 1 has been reorganised according to the comment of reviewer 1
5) On page 8, the sentence “light-emission and photoreception may have probably co-evolved in a signalling context...” in the first paragraph of the section 2. The word “co-evolved” seems too strong expression. The sentence should be rephrased.
The text has been changed and the expression “co-evolved” has been suppressed
6) On page 8, Figure 2 is confusing because light-emission and photoreception seem to be evolved under ambient-light condition. The authors should consider the illustration.
The figure 2 and the legend has been reorganised according to the comment of reviewer 1
7) On page 9, the sentence “It is therefore possible that ...association with the rhizosphere community” in the third paragraph of the section 2.1. In soil, scattering of light seems not to occur. In addition, other luminous organisms (e.g. luminous earthworm) are in soil, and there is a possibility that light from such organisms affects the rhizosphere community or roots of plants. The authors should consider these points.
The text has been changed and improved according to the suggestions and at the end of the paragraph, a sentence discuss the role of bioluminescent earthworms in the soil and the rhizosphere. 3 references (reviews) about bioluminescent earthworms have been added [ref 315-317]
8) On page 10, Figure 3 is confusing because plants seem to indirectly affect photophobic predators. The authors should consider the illustration.
The figure 3 and the legend has been reorganised according to the comment of reviewer 1
9) On page11-13, Table 1 is confusing and not well organized. For example, many empty fields are confusing. The authors should carefully correct the table.
The table 1 has been fully reorganised and improved
Minor points:
1) In the title, the wording “in the unicellular world” seems impressive expression, but may be too strong expression. This reviewer thinks that the wording “in unicellular organisms” might be better.
The title has been changed for “Bioluminescence and photoreception in the unicellular organisms”
2) On page 3, the sentence “The trans to cis photoisomerization of its p-coumaric acid...” in the second paragraph of the section 1.2.1. “p” of the word “p-coumaric acid” should be in Italic. In addition, throughout the paper, the authors should check typos.
The typo has been corrected
3) On page 6, the sentence “this finding also supports the notion that bacteria perceive bioluminescence ... in deep-sea sediments” in the second paragraph of the section 1.2.2. The sentence maybe too strong. This reviewer thinks that the sentence should be toned down a little or rephrased.
The sentence has been toned down: “this finding may also supports the hypothesis that bacteria perceive bioluminescence of other deep-sea organisms, including themselves, which is the unique source of light in deep-sea sediments.”
Reviewer 2 Report
The topic of the review is very interesting, and could have been intriguing for the reader.
Still, this referee has doubts about it. There are barely 10 pages of text, and 439 references, a huge, quite excessive, and disorienting collection of publications, demonstrating the big effort of the authors, which nevertheless is not easily inferred from the way they treated the subject. Most of the information and data about photoreception in aneural organisms are examined superficially, which does prevent a clear understanding of bioluminescence as a communication tool for information exchange.
The feeling is that the review plays the role of a thread linking all the bibliographic entries, without ever provoking questions or stimulating further reading.
Maybe the main difficulty the authors had to face in preparing their review is that many cell functions and photoreceptive mechanisms and behavior in microorganisms are yet to be demonstrated and explained. Howerer, it is difficult to understand their true opinion on the topic, as if they do not dare to discuss the scientific data available up to now.
It is this referee’s opinion that this review it is not relevant, and does not succeed in satisfying the curiosity of the readers as the authors promise in the title.
The rhodopsin in Fig. 1 lacks retinal.
Author Response
Above all, we would like to thank the reviewers for their valuable advice and suggestions to improve the manuscript. We have followed each recommendation regarding text, references and figures. Changes in the revised manuscript are indicated in red.
Reviewer 1
Major points:
1) On page 1, the sentence “Here, we focus our analysis on ... whose bioluminescence is well characterised" in the abstract. The word “characterised” should be changed to the word “studied” because the details of molecular mechanism of dinoflagellate bioluminescence remains to be clarified.
The word « characterised » has been changed by « studied » in the abstract of the revised version
2) On page 2, the sentence “Light-emission is usually produced by ... such as bacteria, dinoflagellates, cnidarian, arthropods and fungi.” in the section 1.1. The photoprotein aequorin should not be treated as one of luciferases. In addition, cited references seem not suitable. At least, the following papers should also be cited. The papers: Nakamura et al., Structure of dinoflagellate luciferin and its enzymatic and nonenzymatic air-oxidation products, JACS 1989, 111, 7607-7611; Wu et al., Solution structure of Gaussia luciferase with five disulfide bonds and identification of a putative coelenterazine binding cavity by heteronuclear NMR, Sci. Rep. 2020, 10, 20069. Furthermore, throughout the Introduction section, the authors should carefully check whether cited references are suitable or not.
The reference and text about aequorin have been suppressed and the proposed references on dinoflagellate luciferin (ref 41) and NMR structure of Gaussia luciferase (ref 48) have been added in the revised version. We have reorganised the text to cite more appropriately the references about bioluminescence. We have added a reference about the verterbrate bioluminescence (ref 37).
3) On page 2, in the second paragraph of the section 1.1. The authors should carefully mention luminous bacteria symbiosis because bioluminescence based on symbiotic bacteria is important as shown in the following review paper. The paper: Tanet et al., Reviews and sysntheses: Bacterial bioluminescence-ecology and impact in the biological carbon pump, Biogeosciences 2020, 17, 3757-3778.
The reference of Tanet et al. has been added (ref 69)
4) On page 4, Figure 1 is confusing. Is the relationship between each luciferase and each photoreceptor clear? What does the color of arrows mean? Photoreceptors and luciferases might be illustrated separately in different panels.
The figure 1 has been reorganised according to the comment of reviewer 1
5) On page 8, the sentence “light-emission and photoreception may have probably co-evolved in a signalling context...” in the first paragraph of the section 2. The word “co-evolved” seems too strong expression. The sentence should be rephrased.
The text has been changed and the expression “co-evolved” has been suppressed
6) On page 8, Figure 2 is confusing because light-emission and photoreception seem to be evolved under ambient-light condition. The authors should consider the illustration.
The figure 2 and the legend has been reorganised according to the comment of reviewer 1
7) On page 9, the sentence “It is therefore possible that ...association with the rhizosphere community” in the third paragraph of the section 2.1. In soil, scattering of light seems not to occur. In addition, other luminous organisms (e.g. luminous earthworm) are in soil, and there is a possibility that light from such organisms affects the rhizosphere community or roots of plants. The authors should consider these points.
The text has been changed and improved according to the suggestions and at the end of the paragraph, a sentence discuss the role of bioluminescent earthworms in the soil and the rhizosphere. 3 references (reviews) about bioluminescent earthworms have been added [ref 315-317]
8) On page 10, Figure 3 is confusing because plants seem to indirectly affect photophobic predators. The authors should consider the illustration.
The figure 3 and the legend has been reorganised according to the comment of reviewer 1
9) On page11-13, Table 1 is confusing and not well organized. For example, many empty fields are confusing. The authors should carefully correct the table.
The table 1 has been fully reorganised and improved
Minor points:
1) In the title, the wording “in the unicellular world” seems impressive expression, but may be too strong expression. This reviewer thinks that the wording “in unicellular organisms” might be better.
The title has been changed for “Bioluminescence and photoreception in the unicellular organisms”
2) On page 3, the sentence “The trans to cis photoisomerization of its p-coumaric acid...” in the second paragraph of the section 1.2.1. “p” of the word “p-coumaric acid” should be in Italic. In addition, throughout the paper, the authors should check typos.
The typo has been corrected
3) On page 6, the sentence “this finding also supports the notion that bacteria perceive bioluminescence ... in deep-sea sediments” in the second paragraph of the section 1.2.2. The sentence maybe too strong. This reviewer thinks that the sentence should be toned down a little or rephrased.
The sentence has been toned down: “this finding may also supports the hypothesis that bacteria perceive bioluminescence of other deep-sea organisms, including themselves, which is the unique source of light in deep-sea sediments.”
Reviewer 2
The topic of the review is very interesting, and could have been intriguing for the reader.
Still, this referee has doubts about it. There are barely 10 pages of text, and 439 references, a huge, quite excessive, and disorienting collection of publications, demonstrating the big effort of the authors, which nevertheless is not easily inferred from the way they treated the subject. Most of the information and data about photoreception in aneural organisms are examined superficially, which does prevent a clear understanding of bioluminescence as a communication tool for information exchange.
The feeling is that the review plays the role of a thread linking all the bibliographic entries, without ever provoking questions or stimulating further reading.
Maybe the main difficulty the authors had to face in preparing their review is that many cell functions and photoreceptive mechanisms and behavior in microorganisms are yet to be demonstrated and explained. Howerer, it is difficult to understand their true opinion on the topic, as if they do not dare to discuss the scientific data available up to now.
It is this referee’s opinion that this review it is not relevant, and does not succeed in satisfying the curiosity of the readers as the authors promise in the title.
We thank reviewer 2 for both his encouragement and criticism.
This paper is indeed new in the field: it seeks to establish a link, in unicellular organisms, between the ability to emit light, photoreception and the ability to integrate stimuli into behavioural responses. Thus, the manuscript cites references from three completely distinct disciplinary fields, which to our knowledge, have never or very rarely been put in parallel. We believe that even in its present imperfect state (as pointed out by reviewer 2), this paper could have the merit of opening up new perspectives in the three disciplines and perhaps stimulate new experiments to verify the proposed hypotheses.
The rhodopsin in Fig. 1 lacks retinal.
We thank reviewer 2 for his keen sense of observation. Indeed, we had forgotten to represent the retinal in the rhodopsin in figure 1. This omission has been repaired in the new Figure 1 of the revised manuscript
Round 2
Reviewer 1 Report
The revision from the authors addresses some of my concerns. However, this reviewer hopes that the authors carefully consider the following major points about figures. Low quality figures will distract readers’ interest in your review. Illustrations by Tanet et al. (the following paper) is one of good examples to improve the authors’ figures. The paper: Tanet et al., Reviews and sysntheses: Bacterial bioluminescence-ecology and impact in the biological carbon pump, Biogeosciences 2020, 17, 3757-3778.
Major points:
1) In panel a of Figure 1, captions in the figure are out of alignment. Parentheses are lacking in the words, Bacteria and Dinoflagellata. In panel b of Figure 1, it is difficult to recognize the name of photoreceptors and the name of co-factors/pigments, separately. “P” in the word “P-coumaric acid” should be lowercase and in Italic. In Figure 4, uppercase is used for designation of the panel, but lowercase is used in Figure 1.
2) In Figure 2, the font size of captions in the figure is too small to read. In addition, what does the size of arrows mean? The authors should carefully consider the illustration.
3) In Figure 3, it is still difficult to distinguish luminous bacteria from non-luminous bacteria although the authors had efforts to improve the illustration. Meaning of colors for bacteria is confusing. Please use color effectively like illustrations shown in the paper by Tanet et al.
4) In Figure 4 B, “Mating”, “Social behaviours”, and “Symbiotic associations” might be illustrated separately in different panels. In Figure 1, lowercase is used for designation of the panel, but uppercase is used in Figure 4.
Author Response
We would like to thank again the reviewer 1 for his help to improve the iconography of the manuscript (we have added this acknowledgements at the end of the manuscript. We have carefully followed all of his advices for improving the figures and legends (in red in the new version)
Reviewer 2 Report
The article I see basic the same of previous version.
i still confirm my doubts. This time no comments, since the previous were not followed.
Author Response
We understand the scepticism of the reviewer 2: our paper proposes indeed totally new perspectives on the relationships between bioluminescence and photoreception in unicellular organisms. However, as stated by Reviewer 2, this work represents a major effort to integrate and synthesize the scientific literature on 3 topics that have, to our knowledge, never been really linked before: (i) bioluminescence, (ii) photoreception and the (iii) cognitive capacity of microorganisms.
It is our double experience in bioluminescence (bacterial and eukaryotic) and our recent works on the properties of molecular networks that process information that led us to explore these new hypotheses.
Our recent work on bacterial bioluminescence (Vannier et al. 2020, ref 60 in the manuscript) has helped to expand the repertoire of bacteria with a lux operon and consequently consider new functions of bacterial bioluminescence. We believe that it is on the basis of this recent work that we have been invited to contribute to this special issue.
On the other hand, the intriguing cognitive properties of unicellular organisms that start to be recognized is really a fascinating topic). Many studies are now converging to support the idea that single-cells can perform highly complex behavioural tasks as illustrated by a special issue of IJMS entitled: From Nanomachine to Nanobrain, Information Processing at a Molecular Scale: https://www.mdpi.com/journal/ijms/special_issues/nano-machines
It therefore seems logical to include the bioluminescence of microorganisms in this emerging conceptual field.
We believe that these new perspectives could be useful to the scientific community and allow the development of new experiments that could either validate or invalidate these hypotheses and perhaps lead to completely unexpected findings.
Round 3
Reviewer 1 Report
The revision from the authors have addressed all of my concerns. This reviewer appreciates the authors’ efforts.
Author Response
Dear Editor,
many thanks for your suggestions that continue to improve the quality of the manuscript. Please find the revised version that includes a conclusion and a different title for table 1.